



# Towards operational multi-GNSS tropospheric products at GFZ Potsdam

Karina Wilgan[1,2,3], Galina Dick[2], Florian Zus[2], and Jens Wickert[2,1]

[1]Technische Universität Berlin, Strasse des 17. Juni 135, 10623 Berlin, Germany
[2]GFZ German Research Centre for Geosciences, Telegrafenberg, 14473 Potsdam, Germany
[3]Wroclaw University of Environmental and Life Sciences, C.K. Norwida 25, 50-375 Wroclaw, Poland

**Correspondence:** Karina Wilgan (wilgan@gfz-potsdam.de)

**Abstract.**

The assimilation of Global Navigation Satellite Systems (GNSS) data has been proven to have a positive impact on the weather forecasts. However, the impact is limited due to the fact that solely the Zenith Total Delays ($ZTD$) or Integrated Water Vapor ($IWV$) derived from the GPS satellite constellation are utilized. Assimilation of more advanced products, such as Slant Total Delays ($STD$s) from more satellite systems may lead to improved forecasts. This study shows a preparation step for

the assimilation, i.e. the analysis of the multi-GNSS tropospheric advanced parameters: $ZTD$s, tropospheric gradients and $STD$s. Three solutions are taken into consideration: GPS-only, GPS/GLONASS (GR) and GPS/GLONASS/Galileo (GRE). The parameters are compared with two global Numerical Weather Models (NWM): European Centre for Medium Weather Forecast (ECMWF) ERA5 reanalysis and a forecast model ICON run by the German Weather Service. The results show that for $ZTD$s and horizontal gradients, all three GNSS solutions show similar level of agreement with the NWM data. For $ZTD$s,

the agreement is better for the ERA5 model with biases of approx. 1.5 mm and standard deviations (SDs) of 7.7 mm than for ICON with biases of 3.2 mm and SDs of 10 mm. For tropospheric gradients, the agreement with both NWMs is very similar: the biases are negligible and SDs equal to approx. 0.4 mm. For the $STD$s, the GPS-only solution has an average bias w.r.t. ERA5 of 4.2 mm with SDs of 25.2 mm. The statistics are very slightly reduced for the GRE solution and further reduced to a bias of 3.5 mm with SDs of 24.5 mm for the Galileo-only observations. This study shows that all systems are of comparable quality.

However, the advantage of combining more GNSS systems in the operational data assimilation is the geometry improvement by adding more observations, especially for low elevation angles.

## 1 Introduction

During the past decades the number of heavy rainfall and other severe weather events has been increasing. One way to improve

the forecasts of such phenomena is to assimilate more meteorological observation data into the Numerical Weather Models (NWMs) (Poli et al., 2007; Zus et al., 2011; Bennitt and Jupp, 2012; Rohm et al., 2019). In addition to the typical data sources for the assimilation such as: radiosonde profiles, satellite and ground-based meteorological observations or aviation data, the Global Navigation Satellite Systems (GNSS) can also provide valuable information. Studies show that the assimilation of the GNSS Zenith Total Delays ($ZTD$s) or Integrated Water Vapor ($IWV$) can have a positive impact on the weather forecasts.



Case-based studies show an increase of the quality of the humidity and precipitation forecasts (Cucurull et al., 2007; Boniface et al., 2009; Kawabata et al., 2013; Saito et al., 2017; Rohm et al., 2019). Nowcasting studies also show an improvement in forecasts, especially for water vapor while using the GNSS estimates (Smith et al., 2000; Benevides et al., 2015; Benjamin et al., 2016).

Some meteorological agencies such as the UK MetOffice, German Weather Service (DWD) or Japan Meteorological Agency
(JMA) are operationally assimilating the GNSS observations. The challenge in the operational assimilation of the GNSS data is that the weather systems are already assimilating many observations from other data sources. Thus, in the related assimilation studies, the GNSS impact is reported just as slightly positive or neutral (Poli et al., 2007; Bennitt and Jupp, 2012; Lindskog et al., 2017). Moreover, these studies are only focused on the use of the tropospheric parameters in zenith direction, i.e. $ZTD$ or $IWV$. More advanced products, such as tropospheric gradients or Slant Total Delays ($STD$s) are of interest, since information
on the horizontal distribution is provided by these parameters. A positive impact of the $STD$ assimilation on forecasts is to be expected, as it provides the tropospheric information in many different directions. The first assimilation experiments of the tropospheric gradients were undertaken by Zus et al. (2019).

This study is conducted within the recent research project Advanced MUlti-GNSS Array for Monitoring Severe Weather Events (AMUSE). The main objectives of this project are: 1) Developments to provide high-quality slant tropospheric delays
instead of only zenith delays, 2) Developments to provide multi-GNSS products instead of GPS-only, 3) Developments to provide ultra-rapid tropospheric information, 4) Monitoring and assimilation of the tropospheric products. Here, we focus on the two first objectives. We show the comparisons of multi-GNSS tropospheric products, obtained using three satellite systems: The US American Global Navigation System (GPS), Russian GLONASS and European Galileo. Since our focus is on the operational assimilation in Germany, for the time-being, we do not use the Chinese Beidou, Japanese QZSS or other
systems. We calculate the tropospheric parameters from three systems for 303 stations in Germany and 613 stations worldwide using the in-house developed software Earth Parameter and Orbit determination System (EPOS8) and compare our estimates with two global NWMs: European Centre for Medium Weather Forecast (ECMWF) ERA5 reanalysis and the forecast model ICON run by DWD. The outcome of this study is a preparation step for the operational assimilation of the $STD$s at DWD. Moreover, the GFZ is one of the analysis centers for the EUMETNET EIG GNSS water vapour programme (E-GVAP[1]) and,
as such, provides the $ZTD$, and in the future $STD$, estimates to the weather agencies for the assimilation.

Many previous studies compared the tropospheric parameters from GNSS and NWM for $ZTD$ or $IWV$ (Vedel et al., 2001; Teke et al., 2011; Wilgan et al., 2015; Douša et al., 2016; Hadaś et al., 2020; Lu et al., 2020; Bosser and Bock, 2021), the tropospheric gradients (Li et al., 2015b; Lu et al., 2016; Douša et al., 2017; Elgered et al., 2019; Kačmařík et al., 2019) or $STD$ (de Haan et al., 2002; Bender et al., 2008; Li et al., 2015a; Kačmařík et al., 2017). However, the majority of these
studies are focused on the comparisons in the zenith directions and the estimates were calculated from the GPS-only data, sometimes GPS/GLONASS combination. This study shows a comprehensive comparison of all three tropospheric parameters, i.e. $ZTD$s, horizontal gradients and $STD$s with a main focus on the multi-GNSS estimates. It is also a first work showing all

---

[1]http://egvap.dmi.dk/



three tropospheric parameters from multi-GNSS solution with fully operational Galileo constellation. A detailed comparison with some selected studies covering similar aspects to this study is shown in section 5 - Discussion.

This introduction section is followed by Section 2 explaining the tropospheric parameters. Section 3 describes the GNSS and NWM data. Section 4 shows the comparison of three different tropospheric parameters, Section 5 discusses our findings in view of the previous studies and Section 6 summarizes our results.

## 2    Tropospheric parameters

The microwave signals propagating through the atmosphere are delayed in its lowest part, the neutral atmosphere, which
consists of troposphere, stratosphere and a part of mesosphere (and is here called 'troposphere' for shortness). The delay is caused by the propagation medium, which is characterized by meteorological parameters: temperature, air pressure and water vapor. The impact can be expressed by the refraction index n. Since this index is very close to unity, usually a parameter called total refractivity $N$ is used (Essen and Froome, 1951):

$$N = 10^6 (n-1). \tag{1}$$

The total refractivity can be calculated from the meteorological parameters using the following equation (Thayer, 1974):

$$N = k_1 \frac{p-e}{T} Z_d^{-1} + k_2 \frac{e}{T} Z_v^{-1} + k_3 \frac{e}{T^2} Z_w^{-1}, \tag{2}$$

where $p$ is the atmospheric air pressure [hPa], $e$ is the water vapor partial pressure [hPa], $T$ is the temperature [K], $k_1 = 77.60$ [K·hPa$^{-1}$], $k_2 = 70.4$ [K·hPa$^{-1}$] and $k_3 = 373900$ [K·hPa$^{-2}$] are the refractivity coefficients, here taken from Bevis et al. (1994); $Z_d^{-1}$ and $Z_w^{-1}$ are the inverse compressibility factors for dry air and water vapor, respectively, usually assumed to
be 1.

From the total refractivity a tropospheric delay in either zenith ($ZTD$) or slant direction ($STD$) can be calculated:

$$\Delta = 10^{-6} \int_S N(s) ds + S - g. \tag{3}$$

where $\Delta$ denotes the delay, $S$ denotes the arc-length of the ray-path and $g$ denotes the geometric distance between the station and the satellite. In the GNSS analysis, the tropospheric delay is approximated according to:

$$STD = MF_h(el) \cdot ZHD + MF_w(el) \cdot ZWD + MF_g(el) [G_N \cos(A) + G_E \sin(A)] + res \tag{4}$$

where $ZHD$ and $ZWD$ are the hydrostatic and wet parts of the $ZTD$, respectively; $G_N$ and $G_E$ denote north-south and east-west gradient components; $MF_h$, $MF_w$ and $MF_g$ are the mapping functions for the hydrostatic, wet part (e.g. Böhm et al.





(2006)) and gradients (e.g. Chen and Herring (1997)), respectively; $el$ is the elevation angle; $A$ the azimuth angle and $res$ are the post-fit phase residuals.

## 3 Data

We have processed the initial data from three multi-GNSS solutions and two NWMs for the month October 2020. In this section, we describe the data sources in more detail.

### 3.1 GNSS data

Our study incorporates GNSS data from three systems: GPS (G), GLONASS (R) and Galileo (E) for 613 stations worldwide from German national network SAPOS, the International GNSS Service (IGS) network, EUREF Permanent Network (EPN) and the GFZ network. Unfortunately, not all used stations are yet adapted to receiving all kinds of the GNSS signals. The number of stations capable of receiving particular signals for the whole world and Germany is given in Table 1. Figure 1 shows the map of all stations for the whole world and Fig. 2 for Germany. For some of our comparisons, we consider only the GRE-capable stations.

**Table 1.** Number of stations capable of receiving particular GNSS signals used in this study.

|         | Whole world | Germany |
| ------- | ----------- | ------- |
| all     | 613         | 303     |
| GRE     | 432         | 218     |
| GR only | 151         | 77      |
| G only  | 30          | 8       |

The data are post-processed with the EPOS8 software developed at GFZ (Dick et al., 2001; Gendt et al., 2004) in the Precise Point Positioning (PPP) mode using the 24 h sliding window approach. The estimates are obtained every hour with 15 minutes sampling rate for $ZTD$/$IWV$/gradients and 2.5 minutes for $STD$s. In the preprocessing step, the GFZ high quality orbits and clocks are estimated using a base of approx. 100 stations located uniformly around the world. The a priori $ZHD$s are taken from the Global Pressure Temperature 2 (GPT2) model (Böhm et al., 2007; Lagler et al., 2013) and the mapping function for the $ZTD$s is the Global Mapping Function (GMF) (Böhm et al., 2006). The mapping function for tropospheric gradients is calculated according to Bar-Sever et al. (1998), i.e. the wet mapping function is multiplied by the cotangent of the respective elevation angle. More processing information can be found in Table 2.

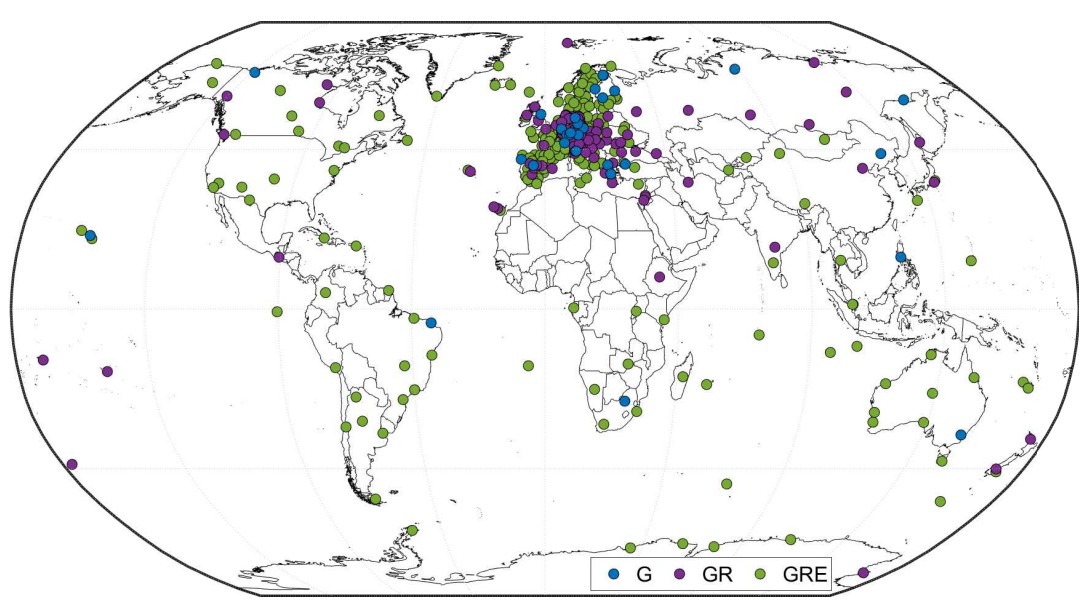

**Figure 1.** Map of all stations used in this study. The colors indicate the capability to receive signals from the particular systems.

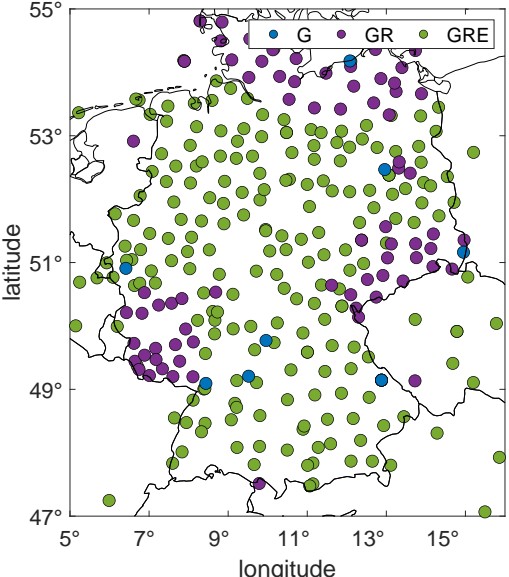

**Figure 2.** Map of the used stations for Germany. The colors indicate the capability to receive signals from the particular systems.



**Table 2.** Characteristics of the multi-GNSS processing at GFZ Potsdam for this study.

| Processing option | Description |
|---|---|
| Observations | Dual-frequency code and phase GPS L1/L2, GLONASS L1/L2 and Galileo E1/E5a observations |
| Products | Precise orbits and Earth rotation parameters calculated using 100 global sites |
| Observations handling | Elevation cut-off angle $7°$, elevation-dependent weighting 1 = cos (zenith > $60°$) |
| | Undifferenced observations with 2.5 min sampling rate |
| Antenna model | IGS08-1854 model (receiver and satellite phase center offsets and variations) |
| Intersystem biases | Estimated as constant, GPS as reference |
| Troposphere | A priori GPT2 model with GMF for $ZTD$ and Bar-Sever MF for gradients |
| | Estimated $ZTD$ and tropospheric gradients every 15 min; $STD$s every 2.5 min |
| | Post-fit residuals applied |
| Ionosphere | Eliminated using ionosphere-free linear combination |
| Loading effects | Atmospheric tidal applied |
| | Hydrostatic loading not applied |
| | Ocean tidal loading applied (FES2004) |
| Gravity | EGM2008 model |

## 3.2 NWM data

The GNSS estimates are compared with the data from two NWMs: the 5th generation reanalysis from ECMWF (ERA5[2]) and

the global forecast Icosahedral Nonhydrostatic (ICON) model run and provided by DWD[3]. The ERA5 data are provided with a horizontal resolution of 0.25° x 0.25° (which translates into around 25 km) on 31 pressure levels. The data is provided with a 3 month delay, however the preliminary data sets are published with a delay of 5 days. The temporal resolution used in this study is 3 h for $ZTD$s and 6 h for $STD$s. There is no ground-based GNSS data assimilation in the model, but the GNSS Radio Occultation (RO) data are assimilated (Healy et al., 2005). For the ICON, the horizontal resolution used in this study is 0.5° x

0.5° with 90 vertical model layers. The temporal resolution is 1 h. The GNSS $ZTD$s and ROs are assimilated into the ICON model.

The refractivity from the NWMs is calculated from the meteorological parameters using Eq. 2 and interpolated from the grid points to arbitrary locations by the methodology described in Zus et al. (2012). The $STD$s for each GNSS satellite-receiver pair are calculated using the in-house developed ray-tracing software described in detail in Zus et al. (2014). The horizontal

gradients from the NWM are calculated by a least-squares adjustment. The used gradient mapping function is the one proposed by Bar-Sever et al. (1998) to match the gradient mapping function that is utilized in the GNSS analysis. The exact description of the methodology of calculating gradients is presented in Zus et al. (2019).

---

[2]https://www.ecmwf.int/en/forecasts/datasets/reanalysis-datasets/era5
[3]https://www.dwd.de/EN/research/weatherforecasting/num_modelling/01_num_weather_prediction_modells/icon_description.html





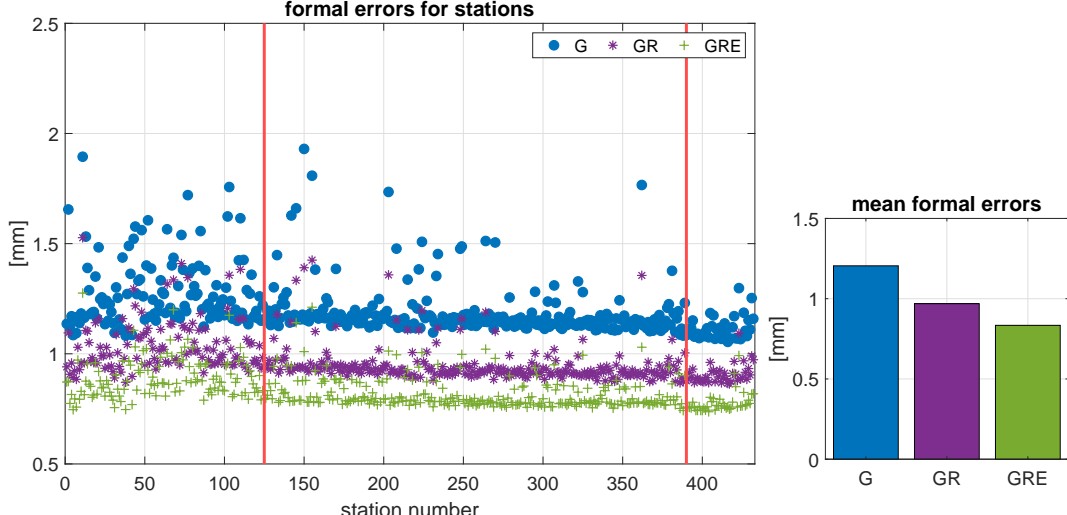

**Figure 3.** Average formal errors for each station in the processing sorted sorted by latitude, southern hemisphere first (left) and the mean formal error averaged from all stations and epochs (right). The red lines indicate the latitude band that includes Germany. The statistics are calculated for October 2020.

## 4 Results

We present the comparison of tropospheric parameters: $ZTD$s, tropospheric gradients and $STD$s obtained from three GNSS
solutions with NWM estimates. We acknowledge that the NWMs are an imperfect reference data source, however, their global coverage makes it convenient to see how the agreement between them and the particular GNSS solutions changes.

### 4.1 Comparisons of Zenith Total Delays

At first, we show the intra-comparisons of the three GNSS solutions and then we compare the solutions with the two NWMs. In the following comparisons, we take into account only the stations that are GRE compatible, i.e. 432 stations for the entire
world and 218 for Germany.

#### 4.1.1 Intra-comparisons of the GNSS solutions

We compare the GNSS estimates from the three solutions, GPS-only (G), GPS/GLONASS (GR) and GPS/GLONASS/Galileo (GRE). At first we take a look at the formal errors of $ZTD$s from the three solutions. Figure 3 shows the errors averaged for each station from the entire month October in 2020 as well as one value for each system, averaged from all the epochs and
stations. We can see that adding GLONASS reduces the formal error from 1.21 mm to 0.97 mm and adding Galileo reduces it further to 0.84 mm.





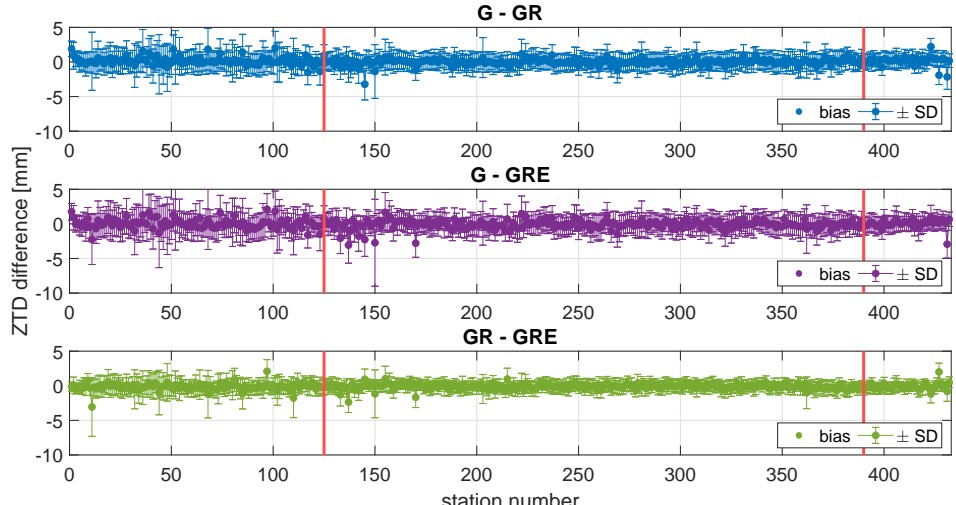

**Figure 4.** The biases and SDs for each station (sorted by latitude, southern hemisphere first) between the three different GNSS solutions. The red lines indicate the latitude band that includes Germany. The statistics are calculated for October 2020.

Figure 4 shows the biases plus/minus their respective standard deviations (SDs) for each station (sorted by latitude, southern hemisphere first) and Table 3 shows the mean biases and SDs averaged from all stations.

**Table 3.** Statistics between the particular GNSS $ZTD$ solutions averaged from all stations for October 2020.

|  | Whole world (432 stations) | | Germany only (218 stations) | |
|---|---|---|---|---|
| Comparison | Bias (mm) | SD (mm) | Bias (mm) | SD (mm) |
| G-GR | 0.07 | 1.54 | -0.04 | 1.42 |
| G-GRE | -0.05 | 1.77 | -0.15 | 1.62 |
| GR-GRE | -0.12 | 1.10 | -0.11 | 0.97 |

Figure 4 shows that the largest differences can be observed for the southern hemisphere, where October is in the spring
season and around the Equator, where the $ZTD$ values are in general larger and more variable. The differences between particular solutions are small. Table 3 shows that the biases are the largest between GR and GRE solutions for the whole world, and between GPS and GRE for Germany. The SDs are the largest between GPS and GRE in both cases.

### 4.1.2 Comparisons with NWM

We compare the three GNSS solutions with the two global NWMs. Table 4 shows the overall statistics of the differences
between each NWM and particular GNSS solutions. Figure 5 shows the biases and SDs for each station for the ERA5 model and Fig. 6 shows the same for the ICON model.





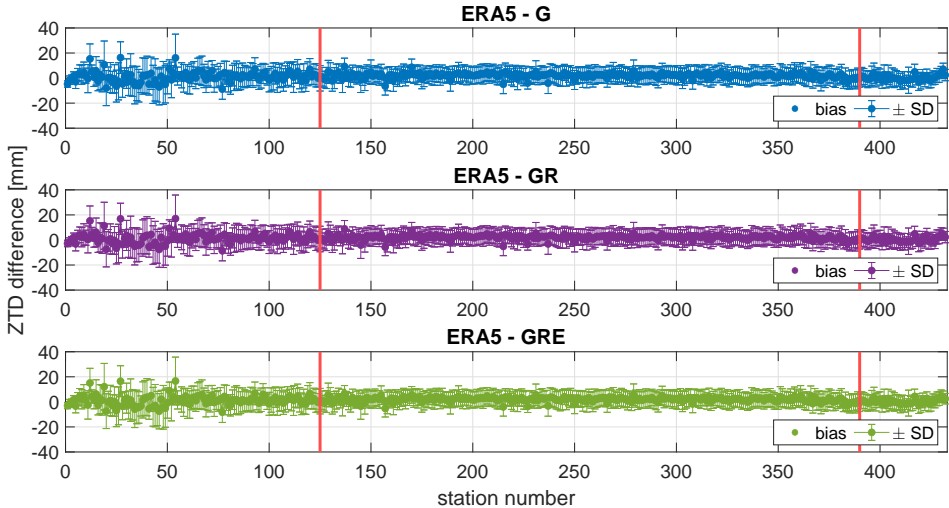

**Figure 5.** The biases and SDs for each station (sorted by latitude, southern hemisphere first) between the ERA5 model and three different GNSS solutions. The red lines indicate the latitude band that includes Germany. The statistics are calculated for October 2020.

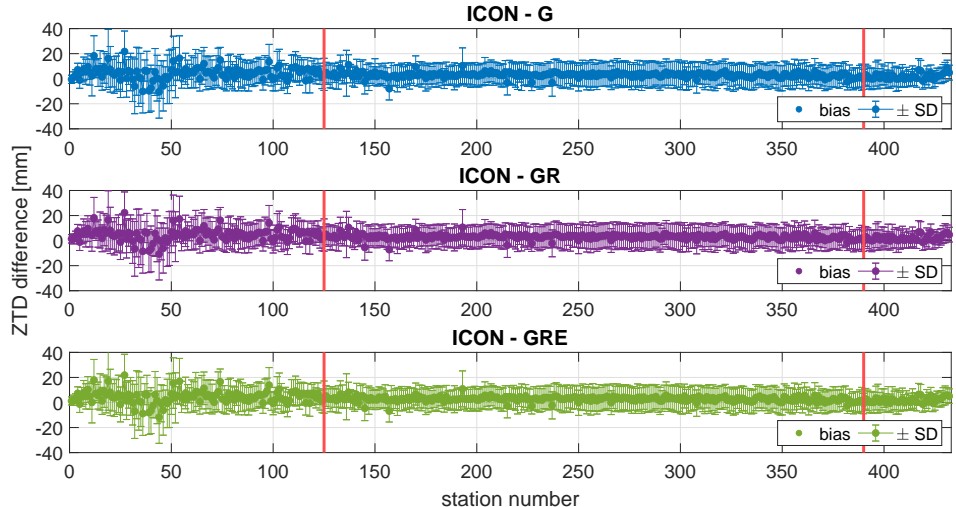

**Figure 6.** The biases and SDs for each station (sorted by latitude, southern hemisphere first) between the ICON model and three different GNSS solutions. The red lines indicate the latitude band that includes Germany. The statistics are calculated for October 2020.





**Table 4.** Statistics between the $ZTD$ NWM and GNSS solutions averaged from October 2020 and all stations.

| | Whole world (432 stations) | | Germany only (218 stations) | |
|---|---|---|---|---|
| Comparison | Bias (mm) | SD (mm) | Bias (mm) | SD (mm) |
| ERA5-G | 1.58 | 7.76 | 2.27 | 6.93 |
| ERA5-GR | 1.64 | 7.73 | 2.25 | 6.90 |
| ERA5-GRE | 1.54 | 7.71 | 2.16 | 6.86 |
| ICON-G | 3.26 | 10.02 | 3.09 | 9.32 |
| ICON-GR | 3.34 | 10.03 | 3.08 | 9.33 |
| ICON-GRE | 3.22 | 10.03 | 2.96 | 9.33 |

The differences between the three GNSS solutions and the NWMs for Germany are very small. For ERA5, the biases and SDs are slightly reduced for GRE compared to GPS-only for the whole world and Germany. For ICON, the biases are very similar for GRE and GPS-only, but larger for GR for the whole world, while for Germany, the biases are reduced for GRE. The SDs are practically the same for all stations. Both NWM models produce too wet conditions (positive biases) w.r.t. the GNSS estimates. While for ERA5 these biases are smaller, on the level of 1.5 mm, they are around 3 mm for ICON. ERA5 is a reanalysis model, while ICON is a forecast model with lower horizontal resolution. Thus, worse statistics for the ICON model are not too surprising. As shown in Figs 5 and 6, we cannot see significant differences between the particular GNSS solutions. Although, we can see that, similar to Fig. 4, there are larger variations for the southern hemisphere and low latitudes. The differences are more pronounced for the ICON model, especially close to the Equator. For Germany (latitudes between 47° and 55°) we cannot see any significant differences between the particular solutions in Figs. 5 and 6. Thus, we take a closer look on the statistics for Germany plotted as a map in Fig. 7 for ERA5 and Fig. 8 for ICON.

Figure 7 shows larger, positive biases for eastern and southern part of Germany, while in the western part they are rather negative. The SDs are almost identical for most of Germany (about 7 mm). The differences between particular solutions are not large, but for some stations both biases and SDs are slightly reduced for the GRE solution. For example, for station OBE4, the biases and SDs from the GRE and GR solutions are significantly reduced compared to the GPS-only solution. There are some stations, especially in the south of Germany, for which the statistics for the GRE solution are reduced. Figure 8, which shows the comparisons for the ICON model, differs significantly from Fig.7. Biases for ICON follow similar pattern as for ERA5, but the magnitude is larger. There is also a visible pattern for the SDs for north and the center of Germany, where the values are of approx. 10 mm. Only for the south of Germany the values are smaller (7-8 mm).

In the following, we present some of the stations with the largest differences between the NWMs and GNSS solutions. Figures 9 and 10 show the $ZTD$ values for the three GNSS solutions and two NWMs as well as the histograms of the residuals for the sample stations: UHOH (Stuttgart, Germany) and OBE4 (Oberpfaffenhofen, Germany), respectively. Please note that the G vs GRE differences and the differences between the biases (ERA5 vs GNSS) are not the same because the ERA5 temporal resolution used for these comparisons is 3 h. For ICON the biases are the same as the differences between solutions, but the SDs are obviously different.



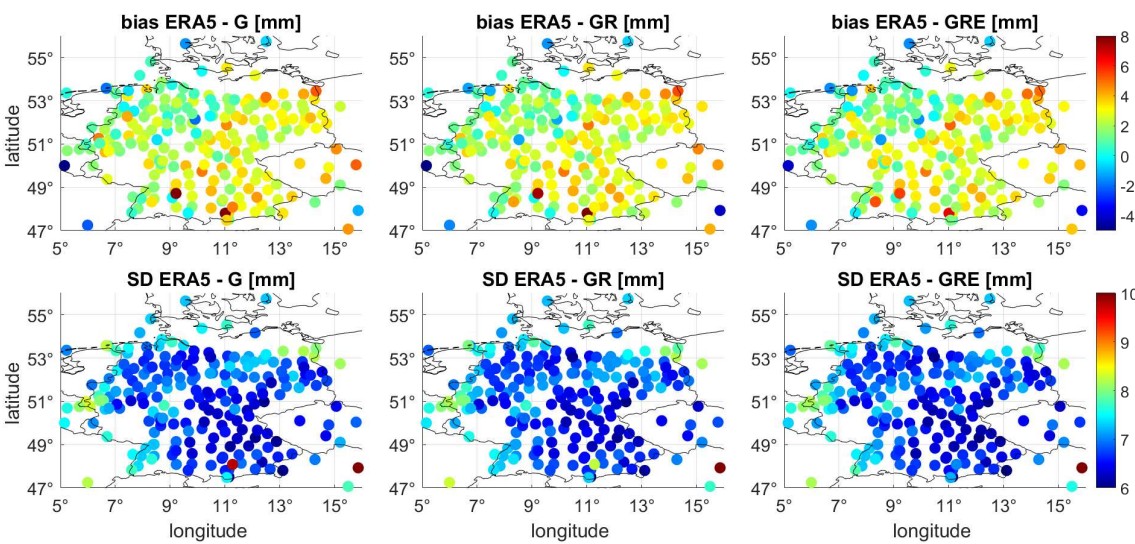

**Figure 7.** Biases and standard deviations between ERA5 and the three GNSS solutions averaged from the month October in 2020 (only for the GRE capable stations, thus there are gaps for some regions).

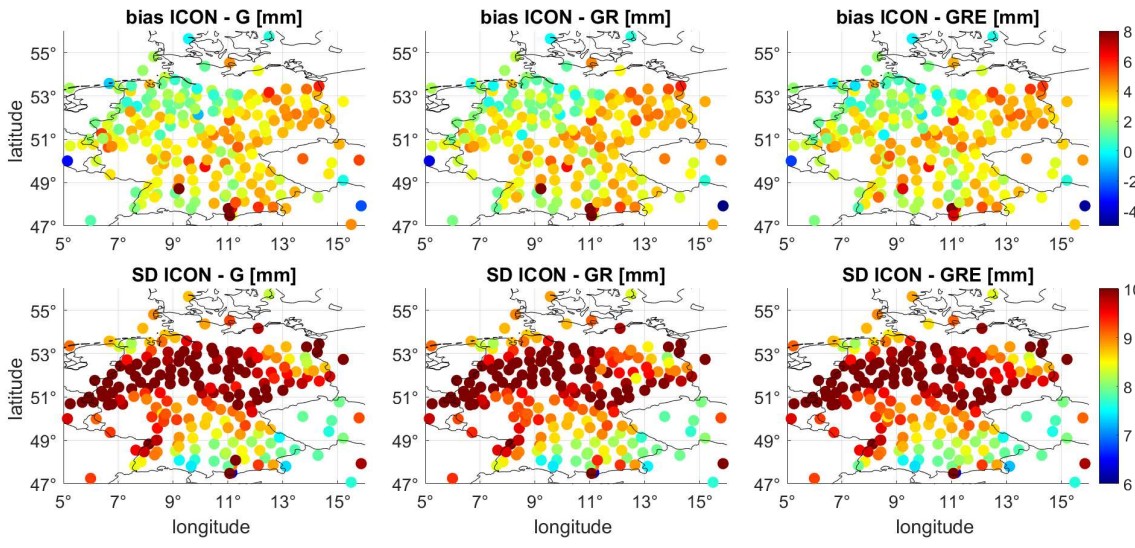

**Figure 8.** Biases and standard deviations between ICON and the three GNSS solutions averaged from the month October in 2020 (only for the GRE capable stations, thus there are gaps for some regions).

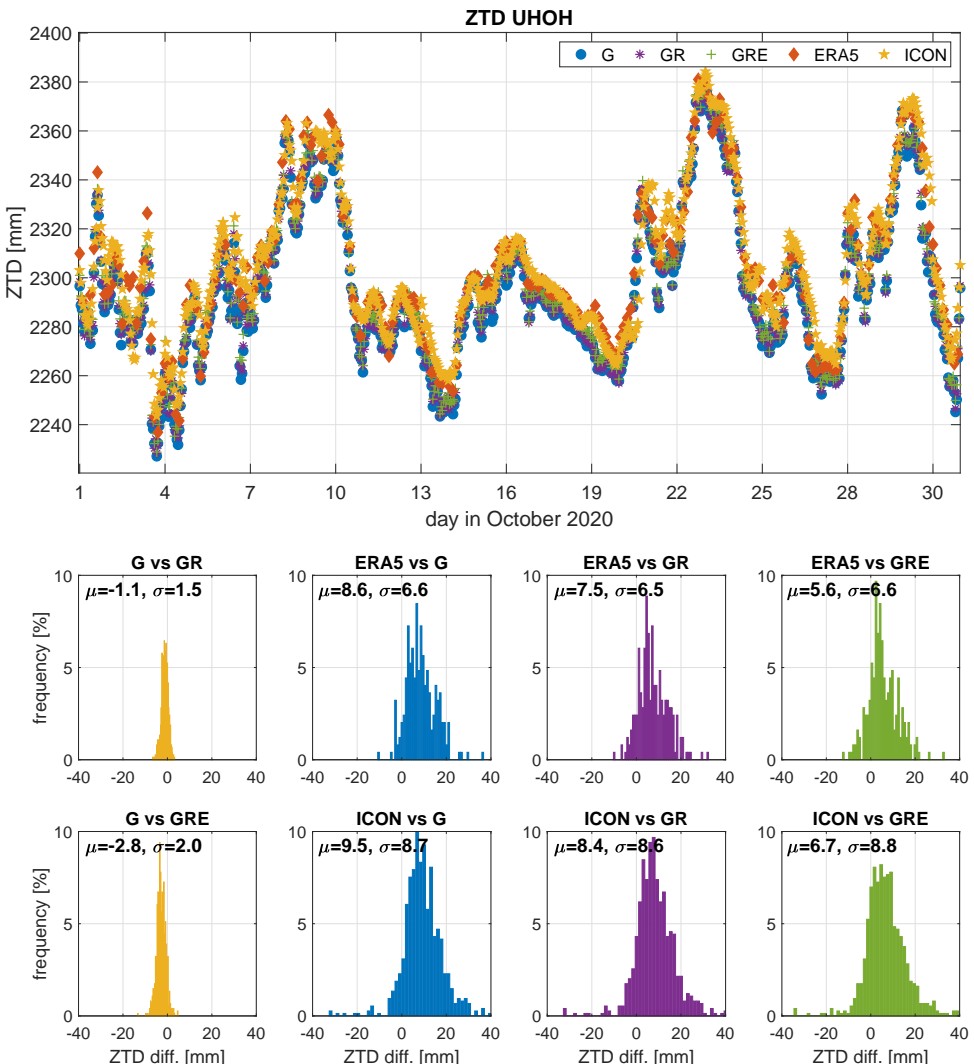

**Figure 9.** The $ZTD$ values for station UHOH (Stuttgart, Germany) from the three GNSS solutions: GPS-only, GR and GRE and two NWM: ERA5 and ICON (top) and histograms of the differences between the particular solutions and models (bottom). The statistics are calculated from October 2020.

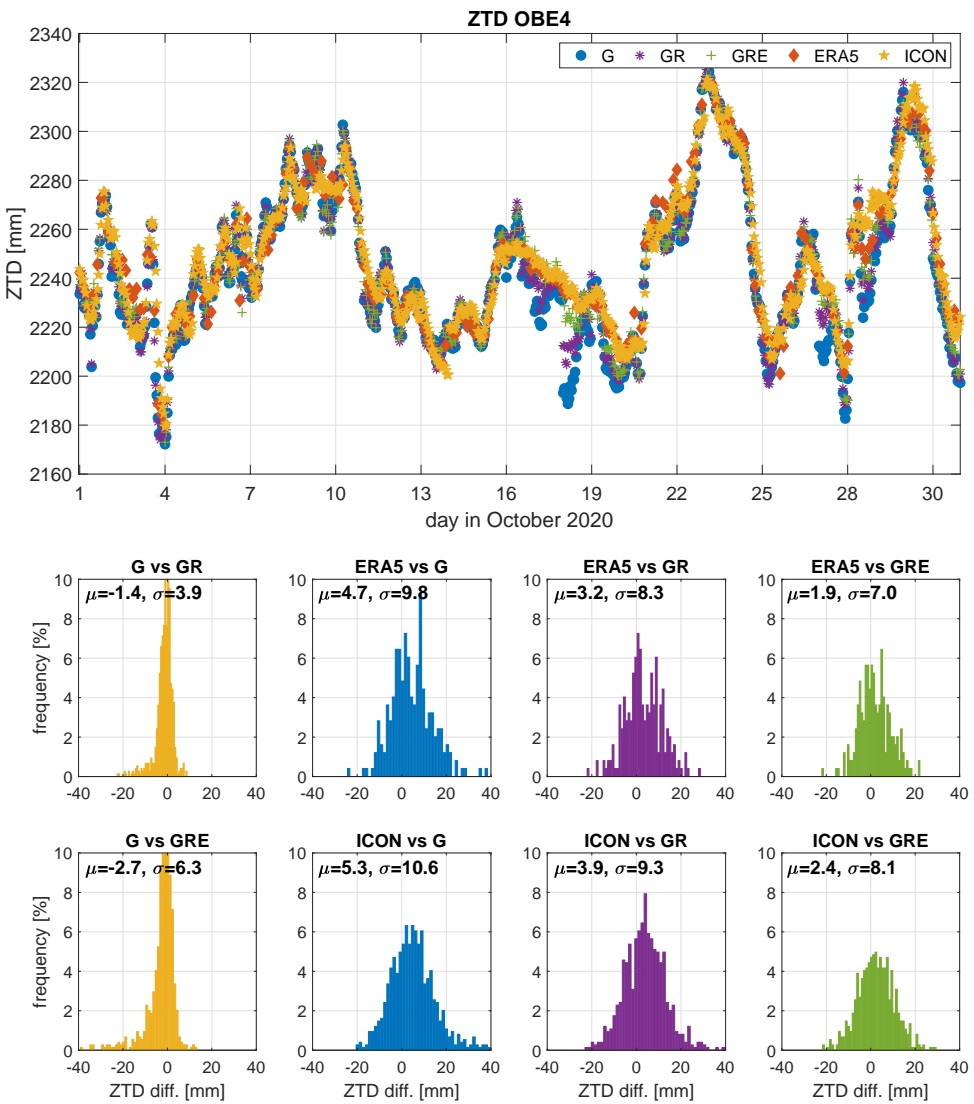

**Figure 10.** The $ZTD$ values for station OBE4 (Oberpfaffenhofen, Germany) from the three GNSS solutions: GPS-only, GR and GRE and two NWM: ERA5 and ICON (top) and histograms of the differences between the particular solutions and models (bottom). The statistics are calculated from October 2020.





Both UHOH and OBE4 have large, positive biases and SDs w.r.t. the NWMs. For the station UHOH (Fig. 9), we can observe a reduction of the biases of around 3 mm for both ERA5 and ICON models, while the SDs remain at the same level. For the station OBE4 (Fig.10) there is a large reduction of the biases and SDs for both models. The comparison with the NWM data shows that for Oct., 19 and Oct., 28 the performances of the GPS-only and GR solutions are much worse than the GRE solution, thus adding Galileo has helped stabilizing the solution.

## 4.2 Comparisons of tropospheric gradients

The tropospheric gradients are a measure of anisotropy in north-south ($G_N$) and east-west ($G_E$) directions. The gradients are of small magnitude, typically below 3 mm. Table 5 shows the biases, SDs and the Pearson's correlation coefficients ($R$) between the three GNSS solutions averaged from all the stations and epochs and Table 6 shows the same statistics but between the two NWMs and three GNSS solutions.

**Table 5.** Biases, SDs and the Pearson's correlations between the three GNSS solutions for tropospheric gradients averaged from October 2020 and all stations.

| Comparison | Whole world (432 stations) | | | Germany only (218 stations) | | |
|---|---|---|---|---|---|---|
| | Bias (mm) | SD (mm) | R (-) | Bias (mm) | SD (mm) | R (-) |
| $G_N$ | | | | | | |
| G-GR | 0.00 | 0.18 | 0.93 | -0.01 | 0.17 | 0.93 |
| G-GRE | 0.00 | 0.21 | 0.90 | 0.00 | 0.20 | 0.90 |
| GR-GRE | 0.00 | 0.12 | 0.96 | 0.00 | 0.11 | 0.97 |
| $G_E$ | | | | | | |
| G-GR | -0.01 | 0.17 | 0.93 | -0.01 | 0.15 | 0.94 |
| G-GRE | -0.01 | 0.20 | 0.89 | -0.01 | 0.18 | 0.91 |
| GR-GRE | 0.00 | 0.13 | 0.96 | 0.00 | 0.11 | 0.96 |

As shown in Table 5, the biases between the particular solutions are very close to zero and SDs are of 0.1-0.2 mm. The largest SDs are between GRE and GPS-only solutions, which was expected. For Germany, the SDs are slightly smaller than for the whole world. The correlations between the solutions are high, around 0.9, and are the highest between GR and GRE solutions and the lowest between GPS-only and GRE.

The values in Table 6 are a few times larger than in Table 5. They may still seem small, but please note that, with the exception of severe weather conditions, the values of gradients are usually below 1 mm. Thus, the SD of around 0.3 mm can actually constitute 30% or more of the entire gradient value. Thus, the differences between NWM and GNSS gradients are considered as significant. The biases are however still rather small. Interesting is the fact that for $G_N$, the biases are always negative, while for $G_E$ always positive. The $G_N$ biases are also a bit larger than the $G_E$ biases. Another remark is that the statistics are very similar for both NWMs, which is not the case for $ZTD$, where the agreement with ICON is worse. Also the differences between the particular GNSS solutions are not pronounced. The correlation coefficients are slightly higher





**Table 6.** Biases, SDs and the Pearson's correlations between the NWM and GNSS solutions for tropospheric gradients averaged from October 2020.

| Comparison | Whole world (432 stations) | | | Germany only (218 stations) | | |
|---|---|---|---|---|---|---|
| | Bias (mm) | SD (mm) | R (-) | Bias (mm) | SD (mm) | R (-) |
| $G_N$ | | | | | | |
| ERA5-G | -0.02 | 0.39 | 0.60 | -0.03 | 0.35 | 0.64 |
| ERA5-GR | -0.03 | 0.39 | 0.61 | -0.05 | 0.35 | 0.65 |
| ERA5-GRE | -0.03 | 0.40 | 0.61 | -0.05 | 0.35 | 0.65 |
| ICON-G | -0.03 | 0.41 | 0.55 | -0.05 | 0.37 | 0.57 |
| ICON-GR | -0.04 | 0.41 | 0.56 | -0.05 | 0.37 | 0.59 |
| ICON-GRE | -0.03 | 0.41 | 0.56 | -0.05 | 0.37 | 0.59 |
| $G_E$ | | | | | | |
| ERA5-G | 0.01 | 0.34 | 0.64 | 0.01 | 0.31 | 0.71 |
| ERA5-GR | 0.00 | 0.35 | 0.65 | 0.00 | 0.32 | 0.72 |
| ERA5-GRE | 0.00 | 0.35 | 0.64 | 0.01 | 0.32 | 0.72 |
| ICON-G | 0.01 | 0.36 | 0.60 | 0.01 | 0.33 | 0.66 |
| ICON-GR | 0.00 | 0.37 | 0.61 | 0.00 | 0.34 | 0.67 |
| ICON-GRE | 0.00 | 0.38 | 0.60 | 0.01 | 0.35 | 0.66 |

for the GRE solution. They are also in general larger for $G_E$ than $G_N$ and also larger for Germany, where the gradients are more consistent. For Germany, the biases are larger than for the entire world, but the SDs are smaller. We do not show the plots analogical to Figs. 4 and 5, but would like to mention, that the statistics (mostly SDs) are also larger for the southern hemisphere and close to the Equator, but magnitude is smaller than for $ZTD$s. To give an example of the gradients' behavior, we plot them for a sample station UHOH. Figure 11 shows the values of the $G_N$ gradient for the station UHOH together with the histograms of the differences. Figure 12 shows the same but for the $G_E$ gradient.

Figures 11 and 12 do not show a visible offset between the NWM and GNSS values like in the case of $ZTD$. The tropospheric gradients, especially from GNSS are much more varying and hard to predict than $ZTD$s. For this particular station (UHOH), there is a reduction of bias for the GRE solution compared to GPS-only solution for both gradient components and both NWMs. For $G_N$, there is also a slight reduction for the SDs, although for $G_E$, the GRE SD is larger than for the other solutions. Moreover, for $G_E$, the biases for the GR solution are larger than for GPS-only. Both gradient components form a vector which points to the local maxima of tropospheric correction, and this usually corresponds to the increasing water vapor content (Douša et al., 2016). To visualize that, Fig. 13 shows gradients for one chosen date, Oct., 29, 12:00 UTC. On that day, a considerable amount of rain, especially in the south-west of Germany was observed (up to 50 mm/day in the southern Bavaria).

The gradients from the NWMs, as shown in Fig. 13, exhibit a clear pattern, pointing to the south-east direction for almost the entire country. They are more consistent with each other than with the GNSS gradients. All the GNSS solutions are very

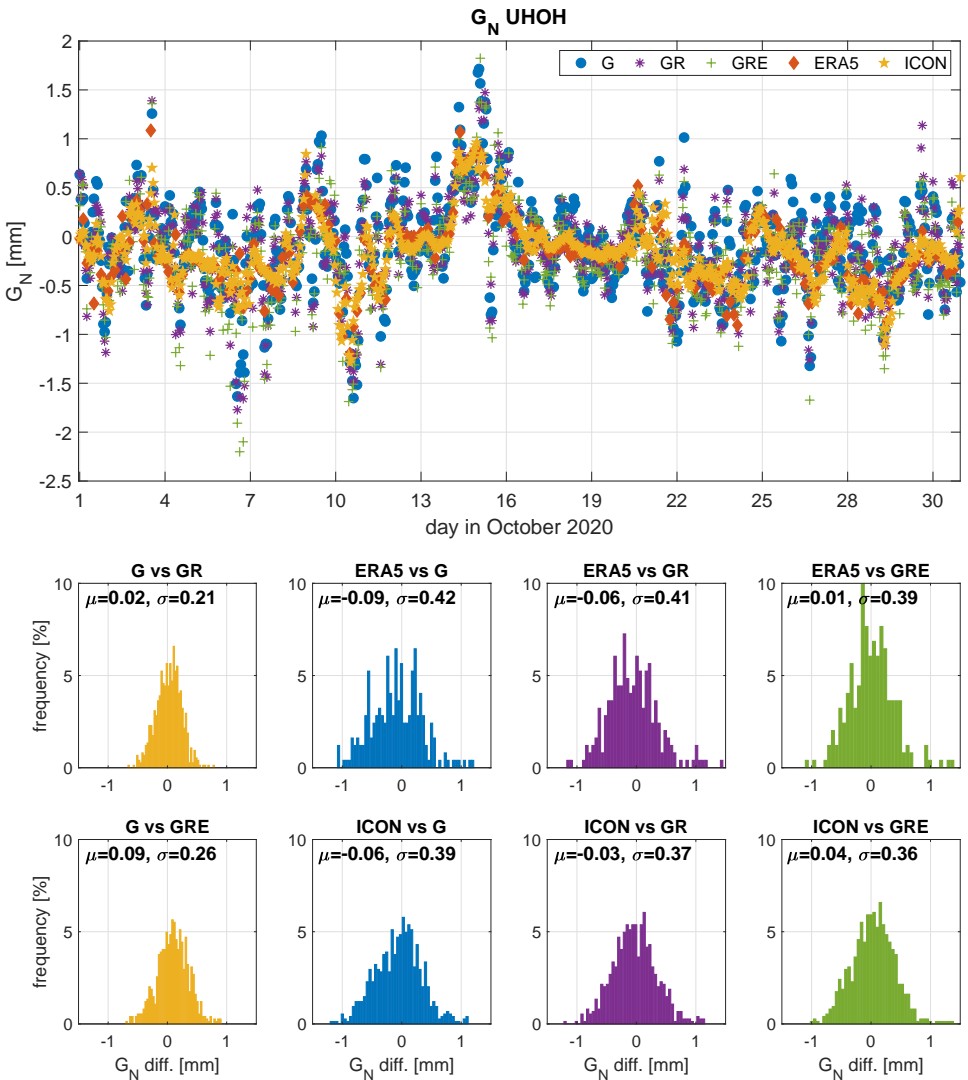

**Figure 11.** The $G_N$ values for the station UHOH (Stuttgart) from the three GNSS solutions: GPS-only, GR and GRE and two NWM: ERA5 and ICON (top) and histograms of the differences between the particular solutions and models (bottom). The statistics were calculated from October 2020.

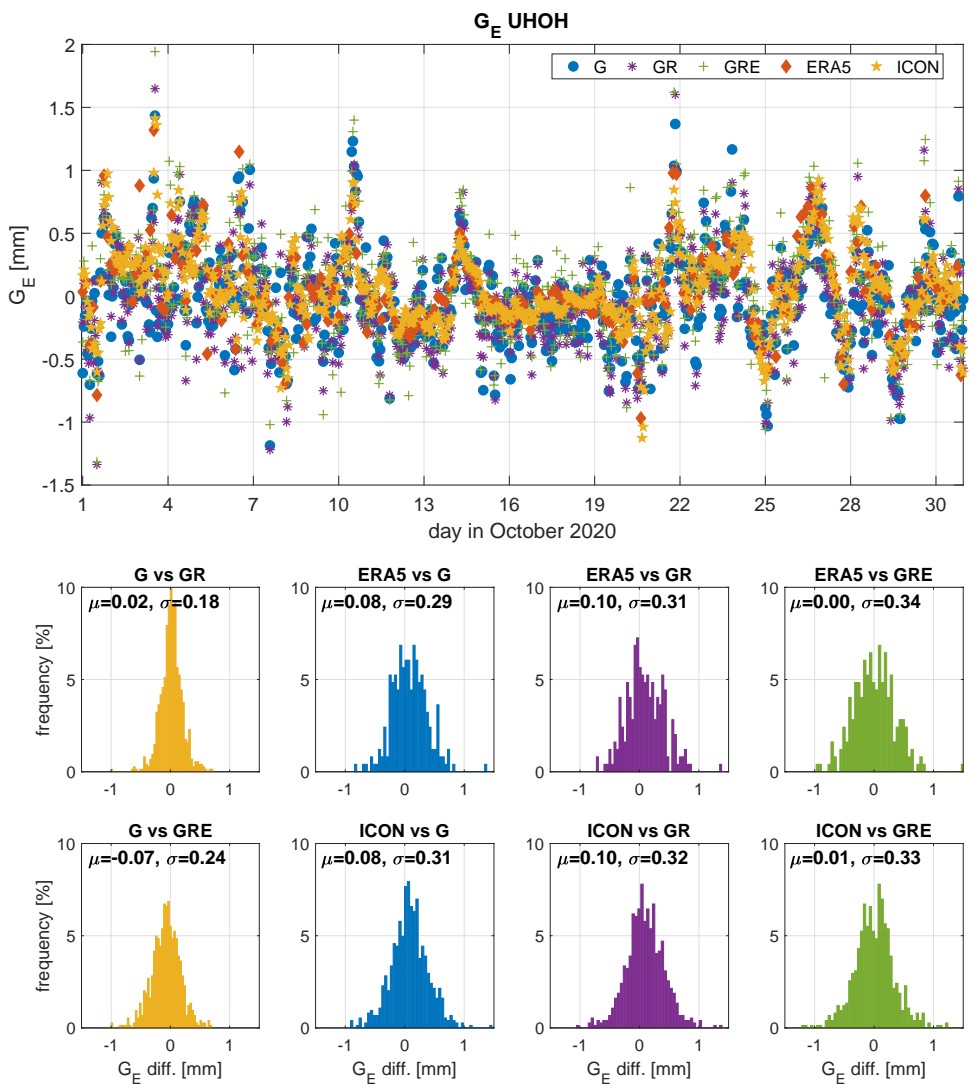

**Figure 12.** The $G_E$ values for the station UHOH (Stuttgart) from the three GNSS solutions: GPS-only, GR and GRE and two NWM: ERA5 and ICON (top) and histograms of the differences between the particular solutions and models (bottom). The statistics were calculated from October 2020.





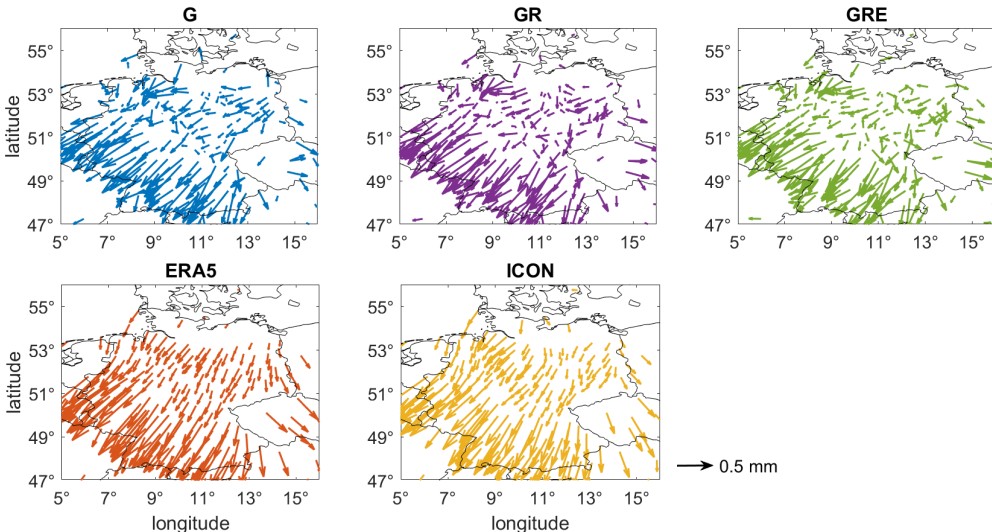

**Figure 13.** The horizontal gradients from the three GNSS solutions and the two NWMs for Germany for a chosen date: Oct, 29, 12:00 UTC.

similar. In general, they also point in the same direction as the NWM gradients, especially in the south-eastern Germany, where
the gradient magnitudes are much larger. The GNSS gradients appear more noisy, especially in the north-eastern Germany. For
this part of the country, the NWM gradients clearly changed direction, but the GNSS gradients do not reconstruct this behavior
so clearly.

### 4.3 Comparisons of Slant Total Delays

From the information in the zenith direction, the tropospheric gradients and the post-fit residuals, the GNSS $STD$s are derived
(Eq. 4). We compare the $STD$s from the three GNSS solutions with the ray-traced $STD$s from ERA5 model. Moreover, we
take the information from all the stations depicted in Figs. 1 and 2 (i.e. 613 stations for the entire world and 303 stations for
Germany), because for $STD$s we have a separate solution for each satellite-station pair, thus there is no need to exclude any
specific stations. Figure 14 shows the differences between the three solutions and the ERA5 estimates for each elevation angle
and the statistics derived from the comparison.
Figure 14 shows larger differences for low elevation angles than close to the zenith. This is due to the fact that the $STD$s for
low elevation angles (here the cut-off angle is $7°$) are around 10 times larger than at zenith. Thus, also the residuals for the low
elevation angles are much larger. We can also see that the number of observations is higher for GRE or GR than for GPS-only,
but the shape of the curves are very similar for all three solutions. The average SDs are also almost identical for all solutions,
although, the biases differ slightly, with the smallest biases obtained from the GRE solution and the largest from GR.
Table 7 shows the statistics for the entire world for the differences between the GNSS solutions and ERA5 model. Due to
the fact that the $STD$ values are much larger for low elevation angles we also show the statistics for the relative differences





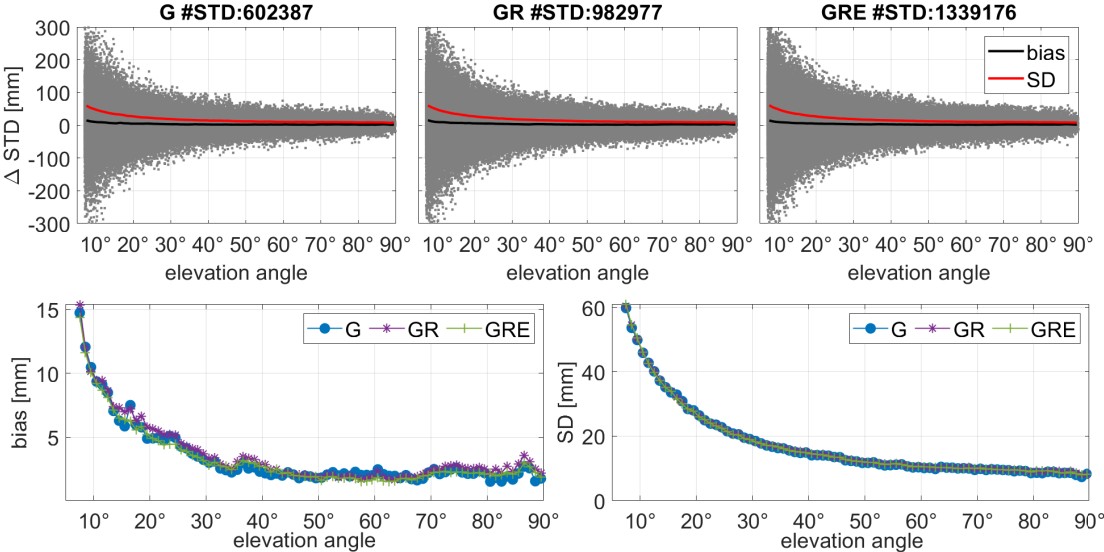

**Figure 14.** Differences between ERA5 and three GNSS solutions for October 2020 for all 613 stations with marked average biases and SDs (top) and the averaged biases and SDs from all solutions altogether (bottom).

($dSTD$s), which are obtained by dividing the differences by the GNSS $STD$ value as well as for the mapped $ZTD$s. These $ZTD$s are calculated using a simple $1/\sin(el)$ mapping function, i.e. $ZTD = \sin(el) \cdot STD$. Table 7 consists also of the statistics for the GPS-, GLONASS- and Galileo-only products, which are extracted from the GRE solution. Table 8 shows the
analogous parameters, but averaged from the German stations.

**Table 7.** Biases and standard deviations between ERA5 and three GNSS solutions (whole world: 613 stations). The statistics are calculated 4 times per day (at 00, 06, 12, 18 UTC).

| comparison | observations | | $STD$ diff. [mm] | | $dSTD$ diff. [%] | | mapped $ZTD$ diff. [mm] | |
|---|---|---|---|---|---|---|---|---|
| | #obs | #outliers | Bias | SD | Bias | SD | Bias | SD |
| G-ERA5 | 602474 | 87 | 4.22 | 25.26 | 0.074 | 0.389 | 1.72 | 9.04 |
| GR-ERA5 | 983135 | 158 | 4.45 | 25.16 | 0.079 | 0.393 | 1.84 | 9.15 |
| GRE-ERA5 | 1339936 | 760 | 4.04 | 24.85 | 0.072 | 0.391 | 1.67 | 9.11 |
| GRE-ERA5 G only | 605052 | 422 | 4.09 | 25.12 | 0.070 | 0.389 | 1.62 | 9.07 |
| GRE-ERA5 R only | 425698 | 262 | 4.29 | 24.69 | 0.078 | 0.394 | 1.81 | 9.18 |
| GRE-ERA5 E only | 309186 | 76 | 3.57 | 24.51 | 0.068 | 0.390 | 1.58 | 9.08 |

As shown in Table 7, the agreement is at a similar level for all solutions. The biases slightly increase for the GR solution, compared to GPS-only or GRE solutions. It is also interesting, that for the GRE solution, if we consider each system separately, the GPS-only product has a better agreement with ERA5 than for GPS-only processing. Moreover, the Galileo-only solution





is better (especially in terms of bias) than any previous solution. The possible reason why the $STD$s from GRE are in higher

agreement with ERA5 than the other solutions, while the $ZTD$s and gradients are at rather similar level is the usage of

the phase post-fit residuals. Due to the Galileo clocks being estimated more precisely than GPS or GLONASS, the post-fit

residuals contain more pure tropospheric information than just noise. Thus, adding them in the calculation of the $STD$s results

in a higher agreement for GRE.

Table 7 also shows the total number of observations and detected outliers calculated using the Chauvenet's criterion. Please

note that due to the coarse temporal resolution of ERA5 and computational costs, the ray-traced $STD$s are calculated only

four times per day. Most of the outliers were found in the GRE solution for GPS observations, even though for the GPS-only

processing there were not that many of them, which shows that processing GPS-only data and extracting the GPS-only data

from the GRE solution results in different data sets.

The biases and especially SDs in Table 7 may appear quite large, but when we calculate the average relative statistics, the

biases from different solutions are around 0.07% and SDs around 0.4%. They are following the same patterns as for absolute

values, i.e. the GRE and especially Galileo-only solution is in a slightly better agreement with ERA5. However, the impact

of the Galileo observations is less visible here compared to the absolute differences. The biases for the mapped $ZTD$s are

very similar to the ones presented in Section 4.1., but the SDs are a bit larger. The possible reason is the usage of the post-fit

residuals which may introduce more noise to the solution. In addition, the usage of the simple $1/\sin(el)$ mapping function may

deteriorate the results (Shehaj et al., 2020).

**Table 8.** Biases and standard deviations between ERA5 and different GNSS solutions (for Germany: 303 stations). The statistics are calculated 4 times per day (at 00, 06, 12, 18 UTC).

| comparison | observations | | $STD$ diff. [mm] | | $dSTD$ diff. [%] | | mapped $ZTD$ diff. [mm] | |
|---|---|---|---|---|---|---|---|---|
| | #obs | #outliers | Bias | SD | Bias | SD | Bias | SD |
| G-ERA5 | 306205 | 0 | 5.93 | 22.82 | 0.099 | 0.336 | 2.34 | 7.86 |
| GR-ERA5 | 507961 | 0 | 5.95 | 22.81 | 0.102 | 0.343 | 2.40 | 8.02 |
| GRE-ERA5 | 688447 | 13 | 5.51 | 22.29 | 0.094 | 0.337 | 2.21 | 7.88 |
| GRE-ERA5 G only | 308622 | 0 | 5.51 | 22.65 | 0.089 | 0.335 | 2.10 | 7.84 |
| GRE-ERA5 R only | 229492 | 0 | 5.46 | 22.25 | 0.097 | 0.343 | 2.28 | 8.04 |
| GRE-ERA5 E only | 150333 | 13 | 5.56 | 21.58 | 0.100 | 0.329 | 2.35 | 7.68 |

For Germany only, as shown in Table 8, we have slightly worse biases than for the whole world (because here the residuals

have mostly the same sign, so the biases do not cancel out), but the SDs are somewhat smaller. There is almost no difference

between the GPS-only solution and GR, but GRE reduces the statistics slightly. For the mapped $ZTD$s, the statistics, especially

the SDs for GR are worse than for GPS-only, even though for $STD$s the values are almost identical. The reason is that for GR

we have more observations for low elevation angles, which being mapped with the simple MF can give larger discrepancies.

Moreover, in Germany, a similar behavior to the whole world can be found for particular systems in the GRE solution, where





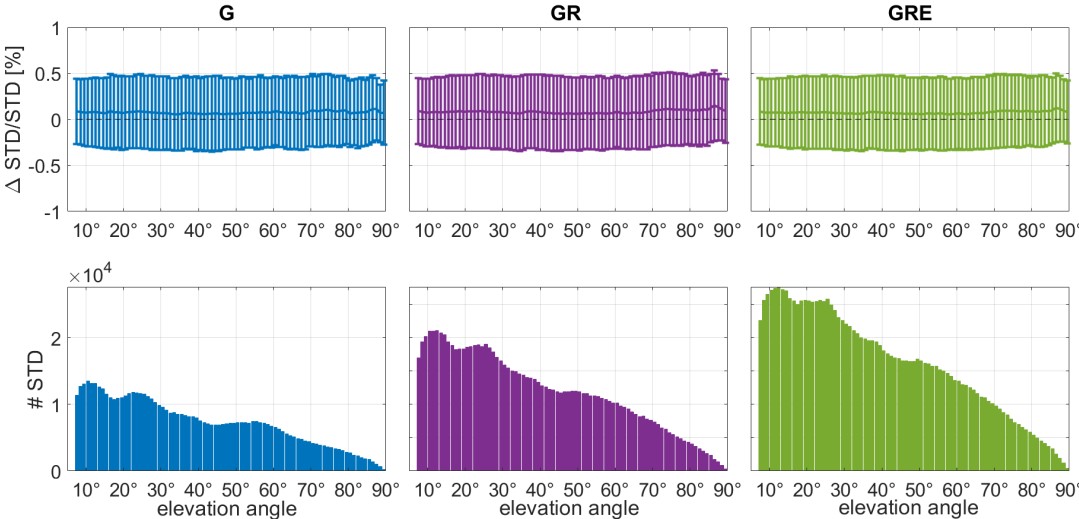

**Figure 15.** Relative differences between ERA5 and the three different GNSS solutions: GPS-only, GR and GRE (top panels) and the number of observations w.r.t. the elevation angle for each solution (bottom).

the SDs for GPS- and GLONASS-only solutions are smaller than in the GPS-only and GR solutions. However, this is not the case for the Galileo-only solution, which has lower SD, but the bias remains at the same level.

Figure 14 shows that the differences between the ERA5 and GNSS estimates depend strongly on the elevation angle. To remove this dependence, we plot in Fig. 15 the relative differences between the model and the GNSS solutions as well as the number of observations for each elevation angle batch.

Figure 15 shows that the relative differences are almost independent from the elevation angle, which means that the solutions are of equal quality for all angles. Only close to zenith, the solutions tend to deteriorate due to the limited number of observations for such angles. The differences between the solutions are rather small as shown in Table 7. Furthermore, one of the advantages of combining the solutions is the increase of the number of observations. Figure 15 shows that adding particular systems increases significantly the number of observations. For this comparison with 6 h resolution, we use over 600,000 GPS, 300,000 GLONASS and 350,000 Galileo observations. Thus, the total number of GRE observations has more than doubled compared to GPS-only observations. What is especially important is that the number of observations for lower elevation angles is increased. For the lowest bin in Fig. 15, there are 11,000 observations for GPS, 17,000 for GR and 22,000 observations for GRE. But also the middle bins are significantly improved, from around 10000 observations for GPS-only to around 25000 for GRE. The $STD$s depend not only on the elevation angle, but also on the azimuth angle of the satellite (see Eq.4). Figure 16 shows the relative differences w.r.t. the azimuth angle and the number of observations for each angle batch.

Figure 16 shows that the relative differences depend slightly on the azimuth angle, especially for the GPS-only solution. However, as shown in the bottom panel, there are only very few observations for azimuth angles close to 0. Adding GLONASS and Galileo observations fills this gap a little and makes the differences less dependent on the azimuth angle. We can conclude



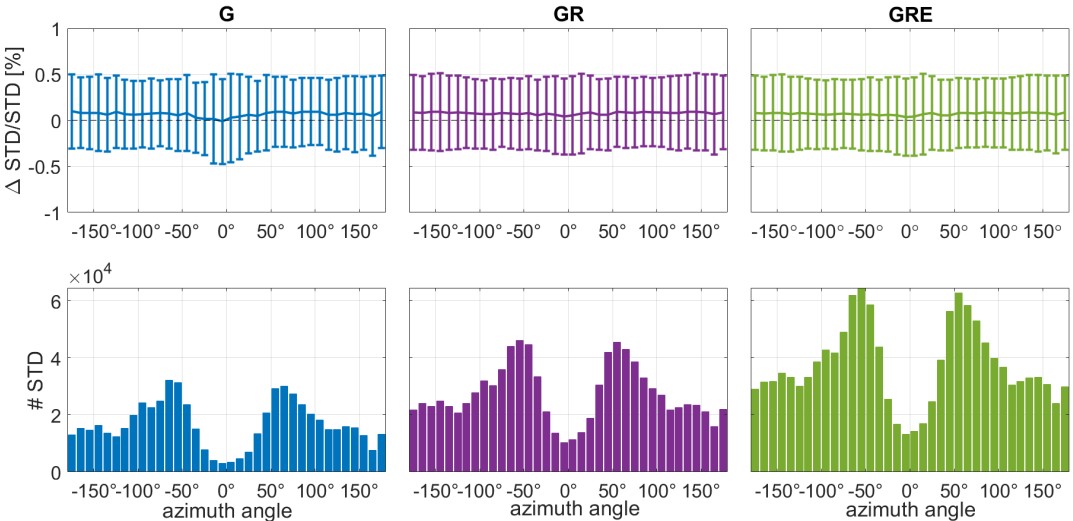

**Figure 16.** Relative differences between ERA5 and the three different GNSS solutions: GPS-only, GR and GRE (top panels) and the number of observations w.r.t. the azimuth angle for each solution (bottom).

that even though adding more systems does not improve significantly the statistics, it increases the number of observations, especially for low elevation and azimuth angles. This addition may lead to more precise information about the tropospheric state obtained via e.g. water vapor tomography.

## 5  Discussion

Comparisons of the GNSS and NWM estimates have already been vastly described in the literature. The majority of the studies focus on the parameters in the zenith direction, either $ZTD$s or $IWV$. Examples have been given in the introduction of this article. Some of these studies have been conducted at GFZ or use the GFZ products. In this section, we would like to summarize a few selected studies and compare our outcomes with theirs.

Douša et al. (2017) compared the tropospheric GPS-only products calculated at 172 stations from almost 20 years of data
(1996-2014) of the second EUREF reprocessing (Repro2). The $ZTD$ comparisons with ERA-Interim reanalysis for almost all variants showed biases of 2 mm and SDs of 8 mm, which corresponds with the findings of this study. For the $G_N$ gradient, the bias was very close to 0 with SD of 0.4 mm and for the $G_E$ gradient, it was -0.05 mm with SD of 0.4 mm. The SDs in this study corresponds with the Repro2 study by Douša et al. (2017), however, our $G_N$ biases are of slightly larger magnitude but the $G_E$ biases are smaller.

Kačmařík et al. (2019) studied different settings of tropospheric gradients for a COST Action ES1206 benchmark period (May-June 2013) for 430 stations in central Europe. The settings included 8 different variants of processing gradients with different mapping functions, elevation cut-off angle, GNSS constellation, observations elevation-dependent weighting and the



processing mode. One of the variants concerned the GPS-only vs the GPS/GLONASS solutions. The comparison with the NWM showed that a small decrease in the SD of the estimated gradients (2%) was observed when using GPS/GLONASS in-

stead of GPS-only. Lu et al. (2016) compared gradients from multi-GNSS solution validated with the ECMWF NWM from 120 stations for three months in 2014. At that time, only 8 Galileo satellites were in use. The results demonstrate that GLONASS gradients achieve comparable accuracy to the GPS gradients, but have slightly more noise and outliers. Compared to the GPS- and GLONASS-only estimates, the correlation for the multi-GNSS processing is improved by about 21.1% and 26.0%, respectively. These studies do not correspond fully with the findings of our study, where the gradients from all three solutions

exhibit a similar level of agreement with NWMs. The reason for higher reduction in these studies and smaller reduction in our study is most probably the usage of a different constraining of the parameters. Kačmařík et al. (2019) and Lu et al. (2016) used loose constraining, while in our study the gradients are more tightly constrained between epochs, but more loose in the general magnitude.

Kačmařík et al. (2017) showed the comparisons of $STD$s from seven different institutions. The authors validated 11 solutions

obtained using five different GNSS processing software packages. They checked different processing strategies, elevation cut-off angle, mapping functions, used products, intervals of calculating the parameters or the usage of post-fit residuals. The tests were performed for 10 reference stations of the COST Action ES1206 benchmark in 2013. This study was restricted to GPS-only and GPS/GLONASS solutions. Amongst the comparisons of many different aspects, it also showed that changing the setting from GPS-only to GPS/GLONASS resulted in a bias of 0.18 mm and SD of 1.95 mm between the solutions, which

is very similar to the current study. GFZ also provided their contribution to the study of Kačmařík et al. (2017), although at that time with a GPS-only solution. This was compared to the NWMs (the GFS and ERA-Interim models). The biases for the mapped $ZTD$s varied for different stations between 4-12 mm with SDs of 7-12 mm for GFS and 0-6 mm with 10-17 mm SDs for ERA5. The agreement is worse than in the current study, probably due to the usage of the data in the warm season, and possibly also due to the different way of calculating the $STD$s from NWM (the assembled and not the ray-traced tropospheric

delays were utilized).

Li et al. (2015a) described real-time comparisons of $ZTD$s, gradients, $STD$s and $IWV$s from 100 globally distributed stations and a 180-day period in 2014 and compared them to the ECMWF operational analysis. In this study, the data from four systems were considered: GPS, GLONASS, Galileo and Beidou (GREC). However, the Galileo data was very limited, there were only 4 satellites in the constellation. Thus, our study is an extension of this previous study with a fully developed Galileo

constellation. The ECMWF vs GREC $ZTD$ comparisons resulted in a fractional bias of 0.1% and SD of 0.5% (corresponding to around 2 and 12 mm), which is a bit worse than in the current study. For gradients (although calculated every 12 h), the authors calculated the root-mean-square error (RMSE), which equaled to 0.34 mm for GREC and 0.38 mm for GPS-only, which was an 11.8% improvement. We do not see such a behavior for our gradients, they are at a similar level regardless of the system. The reason may again be that the gradients from Li et al. (2015a) are very loosely constrained, and this is not the case

of our analysis. For the $STD$s, the authors do not give specific numbers, but visually the GPS-only and GREC solutions are close to each other. The SDs are approx. 1 cm close to the zenith and 10 cm at $7°$, which corresponds with the findings of this paper.





This study is generally in agreement with the findings of the described previous studies. The differences between NWMs and the tropospheric delays, i.e. $ZTD$s and $STD$s are comparable. The main difference concerns the multi-GNSS gradients, which is most likely due to the different ways of constraining the gradient values.

## 6    Summary

This study presented a comparison of tropospheric parameters: $ZTD$s, tropospheric gradients and $STD$s from three GNSS solutions: GPS-only, GPS/GLONASS and GPS/GLONASS/Galileo with two global NWMs: ERA5 and ICON. The three tropospheric parameters using the full Galileo constellation were presented in a publication for the first time. For the $ZTD$s, the formal error was reduced from 1.21 mm for GPS-only solution to 0.84 mm for GRE. Global comparisons with NWMs showed biases of around 1.5 mm with 8 mm SDs for ERA5 and 3 mm with 10 mm SDs for ICON model. This is not surprising as ERA5 is a reanalysis model and ICON is a forecast model. The comparisons for Germany resulted in slightly smaller SDs but larger biases. All three GNSS solutions were very similar. However, there are stations, e.g. UHOH or OBE4, for which adding GLONASS and further Galileo reduced the biases and SDs notably. For tropospheric gradients, the two NWM exhibited very similar behavior, thus the statistics of residuals were also similar for both models. For $G_N$, the average bias was of around 0.03 mm with SDs of 0.4 mm and for $G_E$ 0 bias with 0.3 mm SDs. For Germany, the behavior was similar to the $ZTD$s', i.e. the biases were slightly larger and SDs smaller. For $STD$s, we compared the GNSS estimates only with ERA5. The differences were strongly dependent on the elevation angle, with larger differences for low elevation angles and smaller values close to the zenith. The average bias was around 4 mm with 25 mm SDs which corresponds to 0.07% with 0.4% SDs for the relative values. For $STD$s, adding GLONASS increased the bias but reduced SDs, while adding GLONASS and Galileo reduced both biases and SDs. If we consider only the Galileo observations in the GRE solution, the bias and SD were reduced by 1 mm. For Germany the statistics were again worse for biases and better for SDs, but the GRE solution was still better than the GPS-only and GR by 0.5 mm on average. We analyzed also the relative differences between GNSS and NWM estimates. The dependence on the elevation angle was reduced almost to zero. For the relative differences, the worst agreement was obtained for the values close to the zenith, where there are fewer observations. Moreover, the dependence on the azimuth angle was tested. For the GPS-only solution, where there were only a few observations for angles close to zero, the agreement with ERA5 was worse than for the other angles. Adding GLONASS and Galileo increased also the number of observations for angles close to zero and resulted in better agreement for these angles. In conclusion, the estimates from all three solutions showed a very similar agreement w.r.t. the NWMs. We conclude that they are of similar quality. Nonetheless, adding more systems results in better sky coverage, especially for the low elevation and azimuth angles which leads to a better geometry for the future assimilation and tomography studies.



*Author contributions.* Conceptualization, K.W., G.D, F.Z. and J.W.; methodology, K.W., G.D. and F.Z.; validation, K.W., F.Z.; investigation, K.W., G.D., F.Z.; data curation, G.D.,F.Z.; writing—original draft preparation, K.W. ; writing—review and editing, J.W., F.Z.,G.D.; visualization, K.W., F.Z.; supervision, G.D., J.W.; project administration, J.W.; funding acquisition, J.W., G.D.

*Competing interests.* The authors declare that they have no conflict of interest

*Acknowledgements.* This study was performed under the framework of the Deutsche Forschungsgemeinschaft (DFG) project Advanced MUlti-GNSS Array for Monitoring Severe Weather Events (AMUSE) number 418870484. The authors thank Michael Bender from DWD for providing the ICON data. The ECMWF is providing the ERA5 data. The GNSS observation data are provided by the global networks of IGS, EPN and GFZ. The station and satellite meta-data are taken from the GFZ SEnsor Meta Information SYStem (semisys, Bradke (2020)).





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
