# Peer review of "Towards operational multi-GNSS tropospheric products at GFZ Potsdam"

_Atmospheric Measurement Techniques, 2021_

## Author Comment (AC1)

Dear authors,

Thank you! I enjoyed reading your paper. That said, I sometimes miss clear statement of your novelties and the links/interpretations w.r.t. existing literature. I hope that the questions and suggestions below can help you to improve further your manuscript.

*[KW] Dear reviewer, thank you for your comments. According to the suggestion, we reprocessed the data for the whole year of 2020, instead of just one month. Thus, all the tables with statistics and all the figures in the new manuscript are now changed. By doing so, we had to remove the comparisons with the NWM ICON, as we use this model for fast comparisons and do not store more than a few last months of outputs. However, the main focus of this paper is on the operational multi-GNSS tropospheric parameters retrieval and not on the comparisons with the forecast model ICON. The ERA5 model as a reanalysis is a more suitable reference data source. We also think that by removing the comparisons with ICON, the manuscript became easier to read. We also had to change one of the sample stations from UHOH to POTS, because we only have results for UHOH for half a year.*

*We answer to your comments below. The changes in manuscript are marked in blue.*

**Question 1: (GNSS data processing)**

- In Table 2, you mentioned that the antenna model is IGS08-1854. Why didn't you used a more up-to-date version in IGS14?

*[KW] Thank you for pointing this out. Writing that we use the IGS08 model was an unfortunate copying mistake, we apologize for this. We actually use the IGS14 antenna model. We changed it in the text.*

- Galileo observations: some authors have seen that adding Galileo observations to compute ZTD may introduce a bias (at few mm-level). Have you seen such bias? Are they comparable to amplitude mentioned any existing the literature?

*[KW] We are not certain which bias introduced by Galileo is this question about. In general, the intersystem biases are estimated in the processing (constant per station and per day), with GPS as a reference. But if you mean the ZTD biases as the final product, all presented solutions have similar and adding Galileo is not introducing more bias.*

- Which Galileo satellite PCO/PCVs did you used in your analysis? More recent IGS14 ATX models might include more recent/precise calibrations for Galileo satellites.

*[KW] For the Galileo PCO/PCV we actually use the IGS14 ATX models. We changed it in the manuscript*

- You mentioned that you used atmospheric tidal and hydrostatic loading models. Which models did you used?

*[KW] Hydrostatic loading was not applied while atmospheric S1-S2 pressure loading according to the IERS convention was applied. The information was added.*

- The title of your manuscript state "multi-GNSS" but you have "only" considered the G, R, and E systems. Why haven't you considered others like Beidou? Particularly since you have stations in Asia and you mention that JMA is operationally assimilating GNSS troposphere products.

*[KW] The main focus of this paper is on the Germany where not all stations are capable to track Beidou satellites. Already with Galileo only around half of the stations were able to track Galileo for the entire 2020. We did not want to exclude more stations from the analysis. Moreover, and this is the main reason, Beidou has not yet been implemented in the operational EPOS.P8 software and it requires a lot of work. We however keep this in mind for the future studies.*

- Btw., they don't assimilate "GNSS observations" (Intro, line 29) but "GNSS troposphere products".

*[KW] Corrected*

**Question 2: (short observation period)**

It is well known that the performance of a tropospheric product is not constant over time. Bias and standard deviation will also vary over time and depend on the weather/climatic conditions. How can you conclude about the overall performance of your product based on solely one month of data (Winter, October 2020)? What is the performance of your product during other seasons / under other weather conditions? Including e.g., one year of data or several periods could answer this question.

*[KW] We agree that the statistics depend on the weather conditions (e.g. larger biases and SDs in the warm months), but in terms of agreement of particular solutions, one month of data should be sufficient to show how they differ from each other. Nevertheless, according to the suggestion, we expanded the data period to the whole year 2020. The statistics for the year 2020 are to a great extent in line with the findings for just one month.*

**Question 3: (Dataset sampling and comparison methodology)**

- One major advantage of using ERA5 data is its time resolution of 1h. Your GNSS data processing provides ZTDs every 15 min and STDs every 2.5 min. Why did you compare only at 3 h time resolution for ZTDs and at 6 h resolution for STDs? As you included only one month of data, it can't be a too large number of point (at least for ZTDs and gradients). Also, it seems that for the comparison with ICON, the comparison is done at 1 h time resolution. If yes, why is it not the case for ERA5?

*[KW] We store decades of ERA5 data, thus we keep it in 3 h resolution as it requires a lot of disc space. However, according to the suggestion, we calculated the ERA5 tropospheric parameters with 1 h resolution for the year 2020 for this study. However, we continue to use the 6 h resolution for the STDs, because the ray-tracing through ERA5 model takes a lot of time (especially for one year of data).*

- How did you down-sampled you datasets from 15/2.5 min to 3/6h? Have you only considered simultaneous points or did some kind of average?

*[KW] We only considered simultaneous points.*

- How did you compare GNSS solutions and NWP? Did you choice the nearest grid point from the NWP model or have you implemented something like a weighted mean of the 4 nearest grid points? How did you take into account the altitude difference between the NWP surface model and the GNSS station location? Did you apply any height correction for it? If yes, which one and how?

*[KW] The refractivity at any arbitrary point is obtained by interpolation, i.e., for some arbitrary point (here GNSS station), the four surrounding refractivity profiles are identified. For each refractivity profile, a logarithmic interpolation adjusts the refractivity vertically and then a bilinear interpolation including the vertically adjusted refractivity values is performed. The detailed explanation for the interpolation from NWM is given in the reference Zus et al. (2012). We also added a description to the manuscript.*

**Question 4: (Reconstructing STD using post-fit residuals)**

- When reconstructing your STD, have you compared you results with and without adding the post-fit residuals? Not all studies consistently conclude that post-fit residuals should be added when reconstructing STDs. They usually conclude that post-fit residuals obviously contain some tropospheric information but residuals can also be noisy, hence deteriorating the reconstruction of STDs. This is worth to discuss in your paper!

*[KW] We agree that studies do not conclude clearly on using the post-fit residuals. According to the suggestion, we also added comparison of STDs calculated with and without using post-fit residuals. The results show that the overall agreement with ERA5 for the solution without residuals is slightly better. This is due to the two facts: 1) ERA5 has a sparse horizontal resolution, so it does not resolve well small-scale water vapor 2) residuals contain mostly noise, especially for high elevation angles. However, in cases of severe weather events, there is more tropospheric information in the residuals which may have more positive influence on the NWM assimilation. We added a discussion on this topic to the paper*

- Also, using a more advanced mapping function (rather than just 1/sin(el)) would be better. Have you considered it? What is the estimated impact of using the 1/sin(el) approximation instead of a better mapping function?

*[KW] In the processing we use GMF. The simple 1/sin(el) function is only used to map the STDs back to ZTDs for comparison purposes. It is difficult to interpret statistics for the absolute STDs, as they depend highly on the elevation angle. Thus, we present also the relative statistics and the mapped statistics. And for the mapped ones we use 1/sin(el) for its simplicity. We added a comment about it to the manuscript.*

**Suggestion 1: About the amount of stations / area of interest**

In the introduction, you mention that your area of interest is Germany but processed a global network of 613 stations. Then, the paper alternates/mix results focusing on 218 stations in Germany, other using all stations world-wide, some results on 432 GRE-only stations... This is sometimes a bit confusing. In addition, your results outside Germany clearly emphasize a less good agreement, particularly in the southern hemisphere and low latitudes clearly. If your target is Germany, maybe you can focus on the results of the 218 German station only? This will greatly help in clarity. However, if your targeted area is global (e.g. for data assimilation in global NWP models), then you can still simplify your paper by not mentioning the 613 station but by focusing on the 432 GRE-capable world-wide stations (forgetting to mention about the other 181 station won't affect your findings as your results are based on solely the 432 stations and your are using PPP). Note also that if you retain the world-wide area, your argument of not including Beidou doesn't hold anymore as a significant part of your network is tracking Beidou. Even stations in Europe/Germany does (e.g.,                                                                                                          PTBB00DEU, https://epncb.eu/_networkdata/data_quality/skyplots/index.php?station=PTBB_14234M001).

*[KW] We apologize if the part about the number of stations was confusing. We tried to clarify it in the manuscript. We use only the GRE-capable stations for the comparisons of ZTDs and tropospheric gradients, while for STDs, we can use all stations as we can separate the solutions for each systems. We would like to*

*keep Fig 1 and 2 showing the capability of processing particular stations. We find it interesting that still only around half of the stations is capable of receiving the three systems (for the year 2020 there are less stations capable than just for October 2020, which shows that the stations are being updated).*

*We removed the argument about Beidou from the introduction. However, as explained before, we cannot at the moment include Beidou in our operational software, but we keep it in mind for the future works. We would also like to keep the results for both Germany and the whole world. We find the fact that the agreement is worse for the southern latitudes stations and around Equator interesting. We even added a map of statistics for the entire world (new Fig. 7) for better visualization. Keeping the global stations also makes our results more comparable with the other studies in the 'Discussion' section.*

**Suggestion 2: Merging section 4 (results) and 5 (discussion)**

Merging both sections would increase the readability and quality of the paper by making clear links between your results and the existing literature, by comparing your findings with them, and carrying out an in-depth interpretation. It will also help to emphasize your novelties wr.t. to this literature.

*[KW] Many journals require separating results sections with discussions for greater clarity. Therefore, in this paper we also decided to keep the discussion as a separate section. The studies we discuss were conducted at GFZ or using EPOS.P8 software, thus we can show the advancements of our study compared to theirs. We added some more comments about our current study to this discussion section, so the results can be immediately compared without going back to the Results section. We hope that it will increase the clarity of the comparisons.*

**Suggestion 3: Galileo observations and Outliers**

It seems from your graphs that adding Galileo helps in better solution consistency/stability, i.e., reducing outliers... Developing more this point in your manuscript would be an added value. You can for example analyze the impact of the different solutions on extreme values, outliers... (think e.g., to whisker plots).

*[KW] Thank you for the suggestion. We added box plots of the three solutions showing that GPS-only solution produces more outliers than GR and GRE.*

**Suggestion 4: Improving some plots**

In some plots, you can barely see any differences between the G, GR, GRE time series. Three time series that you cannot differentiate by eye are maybe not worth to show? Have you tried to plot instead the difference between GR and G and between GRE and G (i.e., taking G as a reference)? Would it help your interpretation? For some plots, another trial could be to replace the time series by other metric(s) e.g., a whisker/box plot (that includes outliers) for each 3 solutions?

*[KW] According to the suggestion, we plotted the differences instead of the time-series for some of the plots. The differences are w.r.t. the ERA5 model, since we removed ICON from the analysis.*

**Suggestion 5: ERA5 – ICON agreement**

You can also add an information on how do ERA5 and ICON compare during your studies. That way you will have somehow how "observation compare to model" and "how models compare together".

*[KW] This research mostly focuses on comparing the different GNSS solutions. Due to the reasons explained above, we decided to present only the comparisons with ERA5, thus we will not show the comparisons of ERA5 and ICON.*

**Suggestion 6: Precipitation data**

From your manuscript, it seems you can access precipitation data, at least in Germany. It might be interesting to add this data in your analysis (e.g., aside of ZTD and gradient time series). Does adding more constellation helps estimating ZTD/Gradients/STD during precipitation events? Or does the G, GR, GRE solutions agree together while GNSS solutions and NWP model disagree?

*[KW] We only have access to some precipitation maps from DWD. Thus, we are not able to perform any quantitative comparisons. However, we added a map to Fig 13 to visualize better why the gradients are pointing in the particular direction. The suggestion of checking if more constellations can help during some precipitation/severe weather events is very interesting and we will keep it in mind for future studies.*

**Suggestion 7: Acknowlegements**

Add a reference for IGS and for EPN in the Acknowledgements.

*[KW] Added.*

---

## Author Comment (AC2)

Review of the paper by Wilgan et al: Towards operational multi-GNSS tropospheric products at GFZ Potsdam

The authors compared tropospheric parameters (zenith delays, gradients, slant delays) from different GNSS setups (GPS, GPS+GLONASS, GPS+GLONASS+Galileo) against each other and against parameters derived from numerical weather models (ERA5, ICON). For the first time, the combined all three GNSS and they find good (expected) agreement in their tests. These assessments are carried out globally as well as for GNSS stations in Germany. This is a rigorous comparison, adding new aspects to a long history of studies in that field, which I recommend for publication.

*[KW] Dear reviewer, thank you for your review. We answer to your comments below. The changes in the manuscript are marked in blue.*

I only have minor comments and suggestions:

Please do explain the 24 h sliding window technique with more details. What does it mean to use a 15 min sampling in that case? And a 2.5 min sampling rate for slant total delays?

*[KW]We apologize, but we made a small mistake here. We use the sliding window approach for the operational processing. In this technique, the estimates for each hour are calculated based on the past 24 h of the data and for the next hour we use the previous 23 h plus the new hour, etc. The estimates for this manuscript were post-processed, thus the tropospheric parameters are adjusted using 24 h data intervals (it is more computationally effective and yields similar results). We then removed the information about the sliding window technique from the manuscript. The sampling rates are the same, i.e. 15 minutes for ZTD/IWV/tropospheric gradients and 2.5 minutes for STDs, which means that every hour we calculate 4 values of ZTD/tropospheric gradients and 12 values of STDs.*

I understand that atmosphere non-tidal loading is not applied. Is that correct?

*[KW] Yes, it is correct. Only the tidal loading according to the IERS convention is applied (added in the manuscript).*

Figure caption 3: Average formal error of which parameter? Please add to the caption that this is for ZTD.

*[KW] Added.*

Line 135: I am not sure whether it is correct to say that the ZTD variation is larger close to the equator. I think you can only write that the ZWD is larger. ZHD variations is much larger at higher latitudes.

*[KW] We only left the information that the delays are larger close to the Equator.*

Add to the captions of figure 7 and 8 that these figures are for ZTD.

*[KW] We added the information to all the captions.*

---

## Author Response (AR2)

Dear authors,

thank you for very carefully addressing the comments of the reviewers and by extending your analysis to one year of data instead of one month! You also nicely included some additional topics (adding the post-fit residuals).

[KW] Dear Editor, thank you for the comments. We answer below. The changes are marked in text in blue. Figures 3-6 are replaced with figures with new axes labelling.

I only have some small, technical corrections:
* please add the explanation of the colour coding for your box plots in Fig. 5

[KW] We added the explanation to Fig.5.

* in quite a number of figures (3, 4, 5, 6), the x-axis represent the "station number", and stations are numbered by latitude. Those figures would provide more information if the unit of the x-axis would be latitude directly (with a non-equidistant labeling).

[KW] We replaced Figures 3-6. In the new figures, the labels are referring to the latitude.

* at line 236, you write "Not all studies consistently conclude ...". Please give some references for such studies here.

[KW] We added the references.